# Rootstock Selection for Resisting Cucumber *Fusarium* Wilt in Hainan and Corresponding Transcriptome and Metabolome Analysis

**DOI:** 10.3390/plants14030359

**Published:** 2025-01-24

**Authors:** Lingyu Wang, Qiuxia Yi, Panpan Yu, Sunjeet Kumar, Xuyang Zhang, Chenxi Wu, Zhenglong Weng, Mengyu Xing, Kaisen Huo, Yanli Chen, Guopeng Zhu

**Affiliations:** 1School of Breeding and Multiplication, Sanya lnstitute of Breeding and Multiplication, Hainan University, Sanya 572025, China; wly954692690@163.com (L.W.); wsqx926@163.com (Q.Y.); panpanyu0102@163.com (P.Y.); kumarsunjeet082@gmail.com (S.K.); 15669317226@163.com (X.Z.); chenxii77@163.com (C.W.); w2282072550@163.com (Z.W.); xmy93@163.com (M.X.); 2Key Laboratory for Quality Regulation of Tropical Horticultural Crops of Hainan Province, Tropical Agriculture and Forestry College, Hainan University, Haikou 570228, China; 3Institute of Tropical Bioscience and Biotechnology, Chinese Academy of Tropical Agricultural Sciences, Haikou 571101, China; huokaisen@itbb.org.cn

**Keywords:** rootstock selection, cucumber *Fusarium* wilt, transcriptome analysis, metabolome analysis, WGCNA

## Abstract

Soilborne diseases are important problems in modern agricultural production. *Fusarium oxysporum* f. sp. *cucumerinum* (FOC) is one of the predominant soilborne pathogens threatening cucumber cultivation, especially in Hainan, China. This study assessed FOC-resistant rootstocks using incidence rate, disease severity index (DSI), and area under the disease severity index curve (AUDRC), revealing “JinJiaZhen (Mc-4)” as resistant and “JinGangZhuan 1901 (Mc-18)” as susceptible. Comprehensive transcriptome and metabolome analyses were conducted to investigate the defense mechanisms of these rootstocks, revealing key pathways, such as the mitogen-activated protein kinase (MAPK) signaling pathway, starch and sucrose metabolism, and phenylpropanoid biosynthesis, which are crucial for plant disease resistance. Additionally, the study compared the resistance mechanisms of two other rootstocks, Mc-4 and Mc-18, against FOC infection through transcriptomic and metabolomic analyses. Mc-4 exhibited a higher number of differentially expressed genes (DEGs) related to phenylpropanoid biosynthesis compared to Mc-18. Untargeted metabolomics identified 4093 metabolites, with phenylpropanoid biosynthesis, isoquinoline alkaloid biosynthesis, and porphyrin metabolism as primary annotated pathways. On the sixth day post-inoculation, when the number of DEGs and differentially accumulated metabolites (DAMs) was highest, phenylpropanoid biosynthesis emerged as a key pathway in Mc-4, with 37 DEGs and 8 DAMs identified. Notably, Mc-4 showed upregulated expression of genes encoding enzymes involved in phenylpropanoid biosynthesis and increased accumulation of related metabolites, such as coniferyl-aldehyde, coniferyl alcohol, and coniferyl acetate. These findings highlight the differential defense mechanisms between resistant and sensitive rootstocks and provide insights into plant–pathogen interactions. This study’s results will contribute to the development of better and disease-free cucumber varieties, promoting sustainable agriculture.

## 1. Introduction

Cucumber (*Cucumis sativus* L.) is a versatile vegetable rich in hydration, vitamins, and antioxidants, making it a valuable addition to a healthy diet [1,2]. Its high productivity and demand in the culinary and cosmetic industries makes it an important crop globally, influencing the economy [1,3]. Cucumber is one of the most important winter melon and vegetable crops in Hainan, China. The expansion of Hainan’s cucumber sector is being severely impeded by soilborne diseases like *Fusarium* wilt, which are getting more severe due to years of continuous cropping challenges [4]. *Fusarium* wilt in cucumbers leads to significant yield losses and economic damage by causing severe plant wilting, reduced fruit production, and eventual crop failure [5]. One important strategy to mitigate *Fusarium* wilt’s effect on the cucumber industry is grafting [6]. Grafting in cucumbers enhances their texture and flavor in dishes like salads and tzatziki. This also makes their nutrients more accessible for digestion, and selecting the appropriate rootstock is an important step [7,8,9].

Selecting rootstocks for grafting with the aim of avoiding soil-borne diseases presents a well-studied and widespread challenge. In this regard, Park et al. investigated 65 potential grafting rootstocks to protect melons against *Fusarium oxysporum* f.sp. *melonis* [10]. A similar study used the disease severity index (DSI) to assess rootstock resistance, and eventually four rootstocks were selected [11]. Several other studies on rootstock resistance to *Fusarium* wilt in cucumbers have been investigated [7]. Pavlou et al. used the “disease reaction” metric to choose rootstocks and found that grafting commercial Dutch-type cucumber hybrids onto different resistant Cucurbita rootstocks could be another way to control *Fusarium* wilt [12]. Similarly, Tarek et al. investigated the effect of grafting for cucumber to resist *Fusarium* wilt under heat stress, and highlighted “disease incidence” as the major index for rootstock selection [7]. The genetic resistance mechanism of cucumber to *Fusarium* wilt is complex, potentially involving multiple genes and physiological and biochemical processes. Different regions and different climates will affect the differences in the incidence of *Fusarium* wilt [13] Furthermore, these studies generally lack validation in Hainan, China [14]. Given Hainan’s unique geographical location and climatic conditions that make it an ideal region for cucumber cultivation, this omission not only reduces the applicability of the research findings in this area but also affects their potential for application under similar ecological conditions [14]. Therefore, future research needs to attach greater importance to uncovering disease resistance mechanisms from the molecular perspective and conducting thorough validation in key regions such as Hainan, in order to enhance the practical value and scientific significance of breeding achievements.

Due to innovations in biotechnology, such as transcriptome and metabolome analyses, many new insights have been gained into the advancement of plant sciences [15,16]. Previously, transcriptome analysis has been used to identify important pathways in different cereal crops [17], vegetables, and fruits [18]. More information on transcriptome analysis in plants can be found in comprehensive reviews [19,20]. Wheat resistance to *Fusarium graminearum* has been studied and important signaling pathways and defensive mechanisms highlighted, offering new opportunities for the development of pesticides [21]. Likewise, different metabolomic analyses have been used to study cucumber responses to foliar fertilizers and pesticides [22,23]. Both these techniques have been used for many years to investigate complicated biological responses in plants. Previously, these techniques in combination were used for different plants to investigate complex biological responses. Jiao et al. integrated transcriptome and metabolome analyses to reveal sorghum roots responding to cadmium stress through regulation of the flavonoid biosynthesis pathway [24]. Similarly, Zhang et al. adopted these techniques to reveal the crucial biological pathways involved in rapid adaptive response to salt stress [25].

Although various studies have been undertaken using these transcriptomic and metabolomic approaches, the resistance of grafted rootstocks to cucumber *Fusarium* wilt remains unknown, particularly in Hainan. To overcome this limitation, we designed this study to investigate rootstocks for grafting against *Fusarium oxysporum* f. sp. *cucumerinum* (FOC) in Hainan, China. Given the widespread utilization of *Cucurbita moschata* rootstocks in agricultural practices, this study selected 21 readily accessible melon rootstock varieties from the market as potential germplasm resources. To better evaluate rootstock resistance among 21 grafted pumpkin rootstocks and validate the reliability of our study, we conducted replicated tests and developed a new metric: area under the disease severity index curve (AUDRC). After identifying sensitive and resistant rootstocks, we investigated and examined their responses to FOC inoculation through transcriptome and metabolome analyses.

## 2. Results

### 2.1. Rootstock Selection

Different diagrams of rootstocks inoculated with FOC after 21 days are shown in Figure 1. Incidence rate (IR), disease severity index (DSI), resist disease level (RDL) and area under the disease resistance curve (AUDRC) for different rootstocks inoculated with FOC after 21 days are shown in Table 1. After 21 days of the experiment, the classic disease resistance metric DSI revealed that the rootstocks ShuangLiTieJia (material code 3, Mc-3), JinJiaZhen (material code 4, Mc-4), and ChuangFan 1 (material code 5, Mc-5) significantly outperformed in resisting FOC. These three rootstocks showed better resistance and were assigned a rating of “resistant (R).” The AUCRC can indicate the response speed of rootstocks to FOC and be used to screen better rootstock at the same resist disease level. Among rootstocks Mc-3, Mc-4, and Mc-5 with a resistance level of R, Mc-4 had the lowest AUCRC (Figure 2). Throughout the experiment, rootstocks Mc-3, Mc-4, and Mc-5 demonstrated significant differences in resisting FOC, with rootstock Mc-4 showing the best performance for the three mentioned resistant (R) rootstocks (one-way ANOVA of AUDRC, *p* = 0.02). The rootstocks QuanFuTaiLang (material code CK, Mc-CK), ChuangFan 8 (material code 12, Mc-12), JinGangZhuan 1901 (material code 18, Mc-18), and GenLiShen (material code 19, Mc-19) performed the worst, with a resistance rating of “highly susceptible (HS).” Compared to Mc-4, the AUCRC of CK, Mc-12, Mc-18, and Mc-19 all exhibited significant differences. Taking into account multiple factors, such as the market price and cultivated area of promotion, Mc-18 demonstrated higher research value and potential in the scientific research field. Therefore, we selected Mc-4 and Mc-18 as resistant and sensitive materials, respectively, and conducted further transcriptomic and metabolomic analyses.

### 2.2. Transcriptome Analysis of Resistant and Sensitive Rootstocks

Based on the above results, Mc-4 was selected as the resistant and Mc-18 the sensitive rootstock for transcriptomic analysis. Transcriptomic analysis depicted 5.73 GB of data from each sample, with aligning percentages between 82.5% to 96.3%, and the Q30 value was > 94.3% (Appendix A). A correlation analysis of these FPKM values revealed a consistency of over 90% among the replicates for each sample, indicating that the sequencing’s repeatability is satisfactory. (Figure 3A).

In the comparisons of A0 vs. A2, A0 vs. A4, and A0 vs. A6, Mc-4 exhibited a total of 2763, 1443, and 3312 differentially expressed genes (DEGs) upregulated, respectively, while 2425, 1631, and 4659 DEGs were downregulated in Mc-4 (Figure 3B) (A0 and A2 mean samples were collected from MC-4 at 0 days, 2 days post-inoculation with FOC). The results of Venn diagram analysis showed that there were a total of 837 DEGs upregulated and 876 DEGs downregulated in common across the three comparison groups (Figure 3C,D). In Mc-4, the upregulated DEGs were primarily annotated within metabolism pathways, including phenylpropanoid biosynthesis, biosynthesis of amino acids, carbon metabolism, glycolysis/gluconeogenesis, starch and sucrose metabolism, pentose and glucuronate interconversions, and cyanoamino acid metabolism (Figure 3E). Additionally, the upregulated DEGs were also significantly enriched in processes such as the mitogen-activated protein kinase (MAPK) plant signaling pathway, plant hormone signal transduction, and plant–pathogen interaction (Figure 3E). It is noteworthy that among the downregulated DEGs, plant hormone signal transduction, plant–pathogen interaction, starch and sucrose metabolism, MAPK plant signaling pathway, and phenylpropanoid biosynthesis were also significantly enriched (Figure 3F).

In the comparisons of B0 vs. B2, B0 vs. B4, and B0 vs. B6, Mc-18 exhibited a total of 2067, 2500, and 3340 upregulated DEGs, respectively, while 2498, 3231, and 4499 DEGs were downregulated in Mc-18 (Figure 3B) (B0 and B2 mean samples were collected from MC-4 at 0 days, 2 days post-inoculation with FOC). The results of Venn diagram analysis showed that there were a total of 1140 DEGs upregulated and 1569 DEGs downregulated in common across the three comparison groups (Figure 3G,H). Similarly, in Mc-18, both the upregulated and downregulated DEGs were significantly enriched in biological pathways such as plant hormone signal transduction, the MAPK plant signaling pathway, plant–pathogen interaction, phenylpropanoid biosynthesis, and starch and sucrose metabolism (Figure 3I,J).

After inoculation with FOC, both Mc-4 or Mc-18 were enriched in biological processes such as plant hormone signal transduction, MAPK plant signaling pathway, plant–pathogen interaction, phenylpropanoid biosynthesis, and starch and sucrose metabolism (Figure 3E,F,I,J). This precisely indicates that these play complex and crucial roles in the response of Mc-4 and Mc-18 to FOC. In order to screen out the key factors affecting the resistance of the rootstocks, we selected the sixth-day DEG data for further analysis because the higher number of DEGs on the sixth day emphasizes the significance of determining mechanisms of rootstock resistance. The results showed that the DEGs in A6 vs. B6 were mainly enriched in phenylpropanoid biosynthesis (Appendix A). Additionally, the same results were also observed for both the upregulated and downregulated DEGs in A6 vs. B6 (Figure 3K,L).

### 2.3. Metabolome Analysis of Resistant and Sensitive Rootstocks

A total of 4093 metabolites were identified from Mc-4 and Mc-18 under FOC stress at 0, 2, 4, and 6 days using untargeted metabolomic technology. A PCA showed that PC1 accounted for 24.3%, PC2 accounted for 14.4%, and PC3 accounted for 10.9%, and three replicates of each of the eight groups revealed a high degree of correlation approaching 1 (Figure 4A,B). The results of both the PCA and correlation analysis indicated strong consistency among the three replicates of each of the eight metabolic profile samples, enabling the eight samples to be effectively distinguished and providing robust data support for subsequent analyses. All identified metabolites were annotated using the KEGG database, primarily annotating to biological pathways such as neomycin, kanamycin, and gentamicin biosynthesis (55 species), isoquinoline alkaloid biosynthesis (54 species), porphyrin metabolism (43 species), and phenylpropanoid biosynthesis (29 species) (Figure 4C).

In the comparisons of A0 vs. A2, A0 vs. A4, and A0 vs. A6, Mc-4 exhibited a total of 284, 134, and 118 upregulated differential accumulation metabolites (DAMs), respectively, while 63, 32, and 29 DAMs were downregulated in Mc-4 (Figure 4D). The DAMs in the comparison of A0 vs. A2 were primarily enriched in biological processes such as porphyrin metabolism and the biosynthesis of various plant secondary metabolites. However, the differential metabolites in the comparisons of A0 vs. A4, as well as A0 vs. A6, were not only enriched in porphyrin metabolism but also in significant biological processes like the biosynthesis of isoquinoline alkaloid biosynthesis. (Appendix A–C). In the comparisons of B0 vs. B2, B0 vs. B4, and B0 vs. B6, Mc-18 exhibited a total of 237, 151, and 304 upregulated DAMs, respectively, while 151, 130, and 182 DAMs were downregulated in Mc-18 (Figure 4D). In Mc-18, DAMs from all groups were enriched in the biosynthesis of neomycin, kanamycin, and gentamicin. However, only the DAMs from the comparison of B0 vs. B6 showed significant enrichment in the biosynthesis of isoquinoline alkaloid biosynthesis. (Appendix A–F).

In the comparison between A0 and B0, there were fewer DAMs, and from a metabolic perspective, there was no significant difference between A0 and B0 (Figure 4E). This indirectly suggests that the DAMs of A6 and B6 are caused by FOC. In the comparison of A6 vs. B6, the DAMs were primarily enriched in porphyrin metabolism and isoquinoline alkaloid biosynthesis. This may indicate that porphyrin metabolism and isoquinoline alkaloid biosynthesis are closely related to the disease resistance of Mc-4 and Mc-18.

### 2.4. Integrated Transcriptome and Metabolome Analysis of Resistant and Sensitive Rootstocks

To investigate the differences in resistance to FOC between Mc-4 and Mc-18, we paid particular attention to the A6 vs. B6 comparison group, which exhibited abundant DEGs and DAMs (Figure 3 and Figure 4). During our analysis, we found that the differential genes from the transcriptome data were significantly enriched in phenylpropanoid biosynthesis (Figure 3). Additionally, the metabolome data detected a crucial differential metabolite, 4-hydroxystyrene, which is a product of phenylpropanoid biosynthesis (Appendix A). Based on these findings, we decided to focus our research on phenylpropanoid biosynthesis. In phenylpropanoid biosynthesis, a total of 37 DEGs and 8 DAMs were identified. The coefficient of variation (CV) results between replicate samples for DEGs and DAMs indicated a good degree of consistency in the data across different replicates (Appendix A). The 37 DEGs encode for *phenylalanine ammonia-lyase* (CmoCh07G009540/03G012280/18G009000), *trans-cinnamate 4-monooxygenase* (CmoCh02G006920/20G003310/05G007700/01G007640), p-coumarate 3-hydroxylase (CmoCh16G006230/08G012690), *caffeic acid 3-O-methyltransferase* (CmoCh18G013760/06G006930/06G006980), *4-coumarate-CoA ligase* (CmoCh08G005640/05G006990/08G005650/17G008870), *cinnamoyl-CoA reductase* (CmoCh14G010740/14G010750/09G000170/17G008650/02G012970/01G001900/15G003520/04G028340), *cinnamyl-alcohol dehydrogenase* (CmoCh20G010900/18G005030/20G010910/03G014590/18G004980/11G013930/03G014580), *coniferyl alcohol acyltransferase* (CmoCh14G000800/06G007270/18G012880), and *eugenol synthase* (CmoCh04G028130/09G010220/15G003690) (Figure 5). Compared to Mc-18, the stress imposed by FOC not only upregulated the expression levels of genes encoding enzymes related to phenylpropanoid biosynthesis in A6 but also increased the accumulation of coniferyl aldehyde, coniferyl alcohol, and coniferyl acetate. These results indicate that Mc-4 and Mc-18 may resist the stress imposed by FOC through phenylpropanoid biosynthesis.

### 2.5. Weighted Gene Co-Expression Network Analysis of Mc−4 and Mc−18 Under FOC Stress

A total of 4580 genes (Appendix A) were screened and imported into the WGCNA software package (Version: 1.71) for analysis. The transcriptome data analysis involved performing WGCNA at each epoch. In the hierarchical clustering tree, each branch forms a module, and each leaf in the branches represents a gene (Figure 6A). The dendrogram was then cut into modules (clusters). Identification was based on the correlation between genes (modules) and the relative contents of coniferyl aldehyde, coniferyl alcohol, and coniferyl acetate. As shown in the dendrogram, the WGCNA analysis generated 12 modules, distinguished by their different colors (Figure 6B, Appendix A). Whether the correlation were positive or negative, the magnitude of the correlation indicated the degree of association with the target genes selected from the transcriptome data at that stage (Figure 6B). The modular trait relationships varied across the relative contents of coniferyl aldehyde, coniferyl alcohol, and coniferyl acetate. These modules contained both positively and negatively correlated genes, with expression levels changing under different treatments. We further observed several modules that were highly correlated with the relative content of coniferyl aldehyde, coniferyl alcohol, and coniferyl acetate: MEdarkgreen4 (r = 0.86, *p* = 0.006) in coniferyl aldehyde; MEgreenyellow (r = 0.79, *p* = 0.02) and MEdarkgreen (r = -0.89, *p* = 0.003) in coniferyl alcohol; MEhoneydew1 (r = 0.93, *p* = 8 × 10^−4^) in coniferyl acetate (Figure 6B). Subsequently, we conducted GO and KEGG characteristic gene expression analyses on these four modules. The three most abundant terms in the GO annotations were related to molecular functions, cellular components, and biological processes (Appendix A). The KEGG pathways were enriched in phenylpropanoid biosynthesis and other related processes (Appendix A). Coniferyl aldehyde, coniferyl alcohol, and coniferyl acetate are key substances in phenylpropanoid biosynthesis, which further validates the reliability of our sequencing data.

## 3. Discussion

*Fusarium* wilt is a problem affecting various corps, ornamentals, and garden plants. It drastically affects the yield of cucumber and causes economic loss by causing severe plant wilting, reduced fruit production, and eventual crop failure [26,27]. The resistance of grafted rootstocks to cucumber *Fusarium* wilt remains largely unknown, particularly in Hainan. Therefore, the present study was conducted to investigate rootstocks in Hainan that can resist cucumber *Fusarium* wilt. For this purpose, we initially assessed rootstocks with plant physiological indicators, including incidence rate, disease severity index (DSI), and area under the disease resistance curve (AUDRC). After that, transcriptomic and metabolomic analyses of the resistant and sensitive rootstocks were conducted to highlight the differences between defense mechanisms of FOC.

KEGG enrichment analysis of DEGs in rootstocks revealed that genes in the rootstock were primarily enriched in metabolic pathways, such as phenylpropanoid biosynthesis and plant–pathogen interaction (Figure 3). Upon comparing the transcriptome data of Mc-4 and Mc-18, we observed a significant enrichment of phenylpropanoid biosynthesis in both rootstocks, particularly in the comparison conducted on the sixth day (A6 vs. B6) (Figure 3). This discovery prompted us to focus our research on phenylpropanoid biosynthesis, as this pathway is not only closely associated with plant defense mechanisms but also potentially offers crucial insights into the resistance differences among the rootstocks. Previous studies have found that the phenylpropanoid biosynthetic pathway can contribute to disease resistance in plants [28,29]. Plants have evolved a series of biosynthetic pathways for metabolites, including terpenes, alkaloids, and phenylpropanoids, to adapt to both biotic and abiotic stress [30]. The branches of phenylpropanoid metabolism produce end products such as lignin, flavonoids, hydroxycinnamic acid esters, and hydroxycinnamic acid amides (HCAAs), as well as precursors of lignans and tannins [31]. Studies have shown that lignin can form barriers against microbial infection and herbivory, making it a major contributor to biotic stress resistance [32]. During plant–pathogen interactions, lignin enhances resistance to pathogens, playing a critical role. Lignin is associated with disease resistance, and it increases resistance to enzymatic degradations by increasing mechanical pressure on the cell wall during pathogen invasion, thereby limiting toxins and pathogen invasion [33]. Likewise, a significantly higher content of lignin in resistant plant varieties was found in comparison to susceptible plant varieties, and it increased as the plant maturity increased. Additionally, flavonoids serve as phytoalexins in plant defense [34,35]. Furthermore, resistant varieties that are infected with phytophthora showed an increase in PAL (phenylalanine ammonia-lyase) activity, resulting in higher lignin content. PAL is closely associated with plant–pathogen interactions due to its intermediate phenolic substances and final product (lignin) [36]. Upon pathogen infection, plants can upregulate PAL enzyme activity and lignin content to enhance resistance. Similarly, 4CL (4-coumarate-CoA ligase) catalyzes reactions in the phenylpropanoid pathway in an ATP-dependent manner to form *p*-coumaroyl-CoA and can also catalyze the conjugation of other phenylpropanoid compounds with CoA [37]. However, different 4CL homologues exhibit distinct enzymatic preferences for various phenylpropanoid substrates [38]. Further analysis revealed that upon FOC inoculation, the expression levels of key enzyme genes involved in the phenylpropanoid biosynthesis pathway were upregulated in Mc-4. In contrast, although Mc-18 also exhibited a similar enrichment of this biological pathway, the regulation of gene expression within phenylpropanoid biosynthesis may not be as significant as in Mc-4. This could be attributed to differences in genetic background, physiological characteristics, or metabolic regulatory mechanisms between Mc-4 and Mc-18, leading to their varied resistance levels in response to FOC stress (Figure 5). Additionally, metabolome data analysis demonstrated an increased accumulation of related metabolites such as coniferyl aldehyde, coniferyl alcohol, and coniferyl acetate within the phenylpropanoid biosynthesis pathway. This further indicates that Mc-4 may enhance its resistance to FOC by augmenting the activity of the phenylpropanoid biosynthesis pathway (Figure 5).

The DAMs of resistant rootstock Mc-4 pre- and post-inoculation were enriched in porphyrin metabolism and plant secondary metabolite production (Figure 4C). These metabolites were enriched in plant hormone signal transduction and flavonoid and flavonol synthesis pathways in the early stages of infection. In the later stages of infection, enrichment was predominantly observed in pathways related to isoquinoline alkaloid biosynthesis, ABC transporters, the production of different plant secondary metabolites, and indole alkaloid biosynthesis (Appendix A). This suggests that during the early stages of the disease, Mc-4 improves disease resistance by increasing secondary metabolite production, adjusting plant hormone levels, and increasing flavonoid synthesis, which are associated with plant disease resistance and stress responses [39]. In later stages, Mc-4 strengthens its resistance by generating specific alkaloid compounds with strong antimicrobial properties [40]. Secondary metabolites have an important role in plant defense systems by participating in disease resistance signal transduction and providing a biochemical barrier that effectively hinders pathogen invasion [41]. In addition, with the onset of tobacco mosaic virus (TMV) infection, enhancement of specific proteins in the porphyrin and chlorophyll metabolism pathways can occur. These increases assist in reducing the effects of TMV on tobacco photosynthesis, thereby sustaining the growth and survival requirements of the plant [42]. Furthermore, flavonoids have abundant biological activity and can protect against a variety of diseases. They are varied and widely dispersed in nature [43]. It has been demonstrated that raising a plant’s flavonoid output increases its resistance to infections by regulating the expression of genes involved in the flavonoid biosynthesis pathway [44]. We found that the sensitive rootstock Mc-18 inoculated with *Fusarium* wilt had higher DAMs in pathways linked to the tricarboxylic acid cycle, histidine metabolism, biosynthesis of antibiotics like neomycin, kanamycin, and gentamicin, amino acid metabolism, and lysine degradation. These results suggest that improvement in basic metabolism and production of certain antibiotic compounds can increase disease resistance in Mc-18. These metabolic pathways can provide energy and disease resistance through the synthesis of chemicals and sustainable plant cell metabolism [45]. Similarly, the tricarboxylic acid cycle produces ATP and metabolic intermediates to perform cell functions efficiently [46]. Mc-4 and Mc-18 rootstock have different disease defense mechanisms: Mc-4 focuses on the activation of specific secondary metabolites such as flavonoids and alkaloids, whereas Mc-18 focuses on the improvement of basic metabolism and synthesis of specific antibiotic compounds to effectively control pathogenic organisms.

Regarding the attendance of the pathway “neomycin, kanamycin, and gentamicin biosynthesis” in our metabolome analysis, this phenomenon may indicate a potential involvement of plant defense mechanisms. Neomycin, kanamycin, and gentamicin are aminoglycoside antibiotics, and their biosynthesis involves intermediate metabolites that could play roles in antimicrobial defense in plants. More specifically, during pathogen infection, plants may activate secondary metabolism pathways that produce antimicrobial compounds, either directly inhibiting pathogen growth or enhancing overall resistance. The metabolites related to the biosynthesis of neomycin, kanamycin, and gentamicin could serve similar roles in plant defense. Similar phenomena have been reported in some published studies [47,48].

The main contributions of this paper are in selecting the suitable cucumber grafting rootstock for resisting *Fusarium* wilt in Hainan, China with appropriate quantitative metrics. Transcriptome and metabolome analyses helped us reveal the key signaling pathways and DEGs for rootstocks resisting *Fusarium* wilt, the molecular mechanisms of which may be important for disease control and corresponding pesticide development. This study has provided a detailed analysis of the role of phenylpropanoid biosynthesis in resistance to FOC, but further research is needed to explore the potential contributions of other biological pathways, such as the MAPK plant signaling pathway, porphyrin metabolism, and isoquinoline alkaloid biosynthesis, in conferring resistance to FOC. Additionally, the significant morphological differences observed between Mc-4 and Mc-18 plants at 21 days (Figure 1) suggest that pathways like the MAPK plant signaling pathway and porphyrin metabolism may also play roles in enhancing plant stress tolerance by promoting growth and development. Future studies should investigate these possibilities to gain a more comprehensive understanding of the mechanisms underlying plant resistance to FOC and to identify potential targets for improving crop resilience.

## 4. Materials and Methods

### 4.1. Plant Material and Rootstock Selection

In this study, a total of 21 commercial rootstock varieties derived from *Cucurbita moschata*, including QuanFuTaiLang (Materials Code CK, Mc-CK), were selected as potential germplasm resources [49]. They were sourced from companies such as Shandong Shouguang Vegetable Seed Industry Group Co., Ltd (Weifang City, China). (Appendix A). This field experiment was conducted in November 2022 at the teaching base of the College of Tropical Agriculture and Forestry of Hainan University (longitude: 110.328729, latitude: 20.056729, soil type: latosol). The FOC was provided by the Department of Plant Protection, College of Tropical Agriculture and Forestry, Hainan University. The pathogen was single-cell purified and preserved for further study. Initially, the pathogen was inoculated onto PDA (potato dextrose agar) medium and incubated at 28 °C for 5–7 days. Once the colony fully covered the PDA plate, a sterile punch tool was used in a biosafety cabinet to transfer the fungal disk onto fresh PDA medium. The inoculated plates were then placed on a constant-temperature shaker set to 28 °C and a rotation speed of 200 rpm for 3–4 days to promote fungal propagation. Subsequently, the fungal culture was filtered using a four-layer gauze, pre-sterilized at high temperature, into a beaker. The spore concentration was determined using a microscope and a hemocytometer, and the concentration was adjusted to 1 × 10^7^ cfu.mL^−1^, the required level for inoculation. According to the observation in the preparatory test, an assessment designed to ready individuals for a subsequent, more significant exam or task, the roots of the diseased rootstock were selected for the isolation and identification of the symptomatic fungi. Verification of Koch postulates was completed, symptoms of the purified symptomatic fungi determined, and the highly purified symptomatic fungi stored in a 4 °C refrigerator as the inoculation source of the test. When the seedlings had 2 leaves and 1 heart, inoculation with *Fusarium* wilt was carried out. The root irrigation method of human inoculation was used to inoculate cucumber wilt fungus with a concentration of 1 × 10^7^ cfu.mL^−1^. On the premise of being 1–2 cm away from the taproot and not harming the taproot, a knife was used to cut down from the surface of the soil to injure the roots of the plant and inject 10 mL of the prepared fungus solution with a syringe. The control group was inoculated with the same volume of sterile water. Watering was prohibited for 2 days after inoculation, after which normal management was carried out.

### 4.2. Rootstock Selection Method

After inoculating with the FOC pathogen, the disease incidence level and number of infected plants for each rootstock were recorded every 2 days until the end of 21 days. The incidence rate (IR), disease severity index (DSI), resist-disease level (RDL), and area under the disease severity index curve (AUDRC) for each rootstock were calculated. The formulas for incidence rate, disease index, and area under the disease severity-time curve are as follows:IR (%) = N_d_/N_s_ × 100%,(1)DSI = ∑ (N_r_ × N_v_/N_s_ × N_v-max_) 100%,(2)(3)$$AUDRC=\int_{0}^{T}f\;(t)\;dt,$$
where N_d_ is the number of diseased plants; N_s_ the survey plant number; N_r_ the number of symptomatic plants at all levels; N_v_ the representative values of all levels of symptom severity; N_v-max_ the highest representative value for all disease levels; and f (t) the DSI function over time. The disease severity index was judged based on the extent of visible symptoms on plant tissues, expressed as a percentage to quantify the degree of infection. The disease resistance level (DRL) of a rootstock was categorized as follows: highly resistant (HR) for a DSI value less than 20, resistant (R) for a DSI between 20 and 40, moderately resistant (MR) for a DSI between 40 and 60, susceptible (S) for a DSI between 60 and 80, and highly susceptibile (HS) for a DSI greater than 80. For FOC disease grading and evaluation criteria, refer to Appendix A.

To further investigate the disease resistance mechanisms of rootstocks, samples were collected from MC-4 and Mc-18 at 0 days, 2 days, 4 days, and 6 days (A0, A2, A4, and A6; B0, B2, B4, and B6) post-inoculation with FOC. During sampling, roots from plants of uniform size and growth vigor were selected, with three biological replicates chosen for each time point for transcriptome and metabolome analysis.

### 4.3. Transcriptome Analysis

Samples were collected from MC-4 and Mc-18 at 0, 2, 4, and 6 days post-inoculation with FOC. During sampling, roots of plants with uniform size and growth vigor were selected, and three biological replicates were chosen for each time point. The collected samples were rinsed with clean water, filter paper quickly applied to absorb surface moisture, then the entire roots of the plants were cut off with sterilized scissors and immediately placed in liquid nitrogen for rapid freezing for preservation. The samples were then transferred to an ultralow-temperature freezer at −80 °C for storage in preparation for subsequent testing.

The plant total RNA was extracted using a RNAprep Pure Plant Kit (Tiangen, Beijing, China). RNA concentration and purity was measured using a NanoDrop 2000 (Thermo Fisher Scientific, Wilmington, DE, USA). Per sample, 1 μg RNA was used as input material for the RNA sample preparations. Sequencing libraries were generated using a Hieff NGS Ultima Dual-mode mRNA Library Prep Kit for Illumina (Yeasen Biotechnology (Shanghai) Co., Ltd.) (Shanghai, China) and library quality was assessed on the Agilent Bioanalyzer 2100 system. The libraries were sequenced on an Illumina NovaSeq platform to generate 150 bp paired-end reads. The raw reads were further processed with a bioinformatic pipelinetool, BMKCloud (www.biocloud.net (accessed on 20 October 2023)) online platform. The StringTie reference annotation-based transcript (RABT) assembly method was used to construct and identify both known and novel transcripts from Hisat2 alignment results. Quantification of gene expression levels was estimated by fragments per kilobase of transcript per million fragments mapped (FPKM). Differential expression analysis of two conditions/groups was performed using the DESeq2. The resulting *p* values were adjusted using the Benjamini and Hochberg’s approach for controlling the false-discovery rate. Genes with an adjusted *p*-value < 0.01 and fold change ≥ 2 found by DESeq2 were deemed differentially expressed. A Venn analysis was performed on the DEGs of different categories, and KEGG and GO enrichment analyses were performed on the shared DEGs.

### 4.4. Metabolome Analysis

The LC–MS system for metabolomics analysis was a Waters Acquity I Class PLUS ultrahigh-performance liquid tandem Waters Xevo G2-XS QToF high-resolution mass spectrometer. The column used was purchased from Waters Acquity (UPLC HSS T3 (1.8 um 2.1 × 100 mm)). Negative ion mode: mobile phase A: 0.1% formic acid aqueous solution; mobile phase B: 0.1% formic acid acetonitrile. Injection volume 1 μL. The elution gradient was as follows: 0 min, 98% A/2% B; 0.25 min, 98% A/2% B; 10.0 min, 2% A/98% B; 13.0 min, 2% A/98% B; 13.1 min, 98% A/2% B; 15.0 min, 98% A/2% B. The parameters of the ESI ion source were as follows: capillary voltage: 2000 V (positive ion mode) or −1500 V (negative ion mode); cone voltage: 30 V; ion source temperature: 150 °C; desolvent gas temperature 500 °C; backflush gas flow rate: 50 L/h; desolventizing gas flow rate: 800 L/h. The raw data collected using MassLynx V4.2 were processed by Progenesis QI software (Version: 2.4) for peak extraction, peak alignment, and other data processing operations, based on the Progenesis QI software online METLIN database and Biomark’s self-built library for identification. Theoretical fragment identification and mass deviation All were within 100 ppm. After normalizing the original peak area information to the total peak area, follow-up analysis was performed. Principal component analysis and Spearman correlation analysis were used to judge the repeatability of the samples within groups and the quality control samples. The identified compounds were searched for classification and pathway information on the KEGG, HMDB and Lipid Maps databases. T tests were used to calculate the significance (*p* value) for each compound. The R language package ropls was used to perform OPLS-DA modeling, and 200 permutation tests were performed to verify the reliability of the model. The VIP value of the model was calculated using multiple cross-validation. The method of combining the difference multiple, the *p* value, and the VIP value of the OPLS-DA model was adopted to screen the differential metabolites. The screening criteria were FC > 1, *p* value < 0.01 and VIP > 1. The difference metabolites of KEGG pathway enrichment significance were calculated using hypergeometric distribution test.

### 4.5. WGCNA Co-Expression Network Construction

Using the WGCNA expression data of Mc-4 and Mc-18 under FOC stress, a co-expression network was constructed using the WGCNA package in R software version 4.3.3 [50]. The relative content of coniferyl aldehyde, coniferyl alcohol, and coniferyl acetate in Mc-4 and Mc-18 under FOC stress were imported as trait files. The results were visualized using Cytoscape software version 3.7.1 [51].

### 4.6. Integrated Transcriptome and Metabolome Analysis

A comprehensive joint analysis of the transcriptome and metabolome was conducted to uncover the vital metabolic pathways involved in Mc-4 and its response to FOC stress, with detailed evaluation of the alterations of key metabolites and genes associated with the enzymes crucial to these pathways. The DEGs and DAMs in the phenylpropanoid biosynthesis pathway between A6 and B6 were normalized and visualized using TBtools version 1.098 and subsequently illustrated using Adobe Illustrator 2022 software.

### 4.7. Statistical Analysis

Data were collated using WPS Office 12.1.0.17827 software and GraphPad 8.0.2.263 software for one-way ANOVAs, and the least significant difference (LSD) method was used for multiple comparisons, with significant differences defined as *p* < 0.05. All data are presented as the means ± standard errors of three replicates.

## 5. Conclusions

In this paper, we investigated rootstock selection for resisting *Fusarium* cucumber wilt in Hainan. The results identified the suitable rootstock for resisting FOC and highlighted potential key DEGs and pathways to assist in resisting FOC. Finally, differences in defense mechanisms for resistant and sensitive rootstocks were validated by metabolomic enrichment analyses.

## Figures and Tables

**Figure 1 plants-14-00359-f001:**
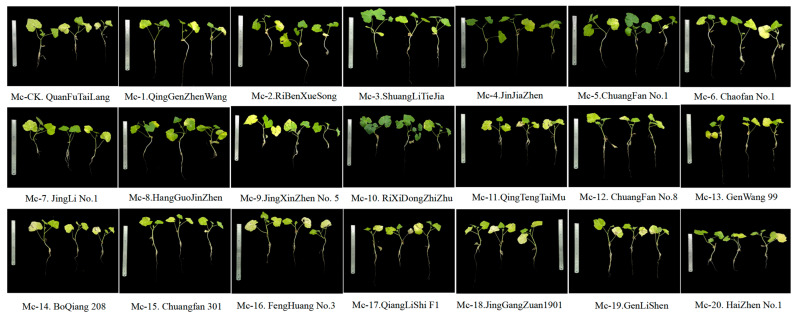
Photographs of rootstocks inoculated with FOC after 21 days. As can be seen from the figure, the rootstock JinJiaZhen exhibits better disease resistance compared with other rootstocks in some degree. Even at 21 days post-inoculation, this variety maintains a healthy state with relatively green leaves, indicating its superior resistance to *Fusarium* wilt. This contrast is quite evident when compared with other rootstocks that are markedly affected by the pathogen, such as QuanFuTaiLang and GenLiShen, where the plants display obvious wilting symptoms.

**Figure 2 plants-14-00359-f002:**
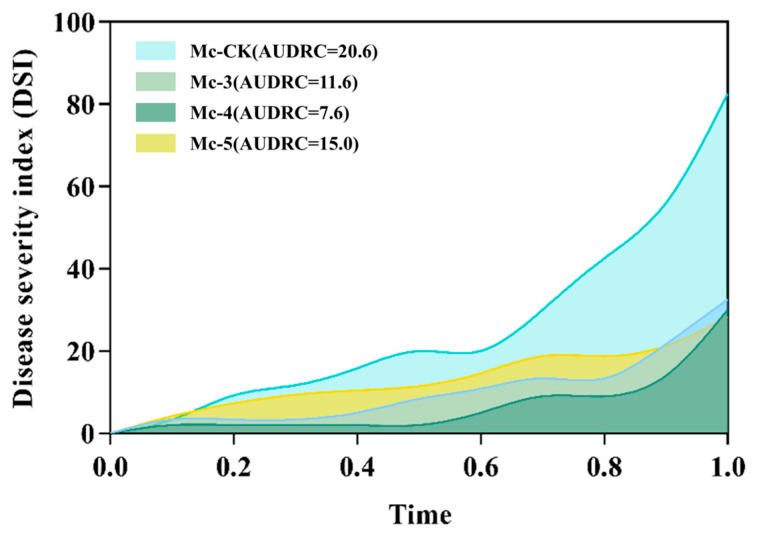
AUDRC results of rootstocks with resistance level of “resistant (R)”. The X-axis in Figure 2 represents the time elapsed since the inoculation of the rootstocks. The 21st day of the experiment is normalized to 1 on the X-axis, while the first day is normalized to 0. For reference, the 5th day is represented as 0.2 on the X-axis. This time transformation was introduced to align with the definition of the area under the receiver operating characteristic (AUC) curve in machine learning, ensuring that the highest AUC value is standardized to 1. This normalization will facilitate future comparisons between different experiments.

**Figure 3 plants-14-00359-f003:**
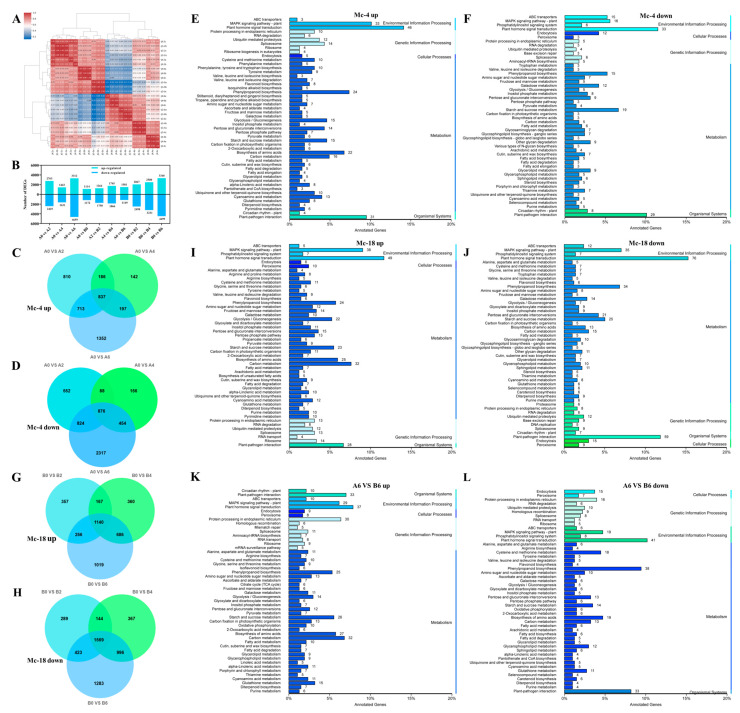
Analysis of transcriptional differences between Mc-4 and Mc-18 under FOC stress conditions. (**A**) Transcriptome correlation analysis under FOC stress. (**B**) Changes in DEGs in Mc-4 and Mc-18 under FOC stress. “Up” represents upregulated genes, and “down” represents downregulated genes. (**C**,**D**,**G**,**H**) Venn analysis ((**C**) Mc-4 upregulated differential genes; (**D**) Mc-4 downregulated differential genes; (**G**) Mc-18 upregulated differential genes; (**H**) Mc-18 downregulated differential genes). (**E**,**F**,**I**–**L**) KEGG pathways of differentially expressed genes ((**E**) Mc-4 upregulated differential genes; (**F**) Mc-4 downregulated differential genes; (**I**) Mc-18 upregulated differential genes; (**J**) Mc-18 downregulated differential genes; (**K**) A6 vs. B6 upregulated differential genes; (**L**) A6 vs. B6 downregulated differential genes). A0, and B0: Mc-4 and Mc-18, which were not inoculated with FOC, served as the control group. A2, A4, and A6: Mc-4 was treated with FOC for 2, 4, and 6 days. B2, B4, and B6: Mc-18 was treated with FOC for 2, 4, and 6 days. The length of the bars in the bar chart and the numbers on the right side indicate the number of DEGs between groups. A higher number of DEGs suggests that the pathway may be an important contributor to the differences observed between groups.

**Figure 4 plants-14-00359-f004:**
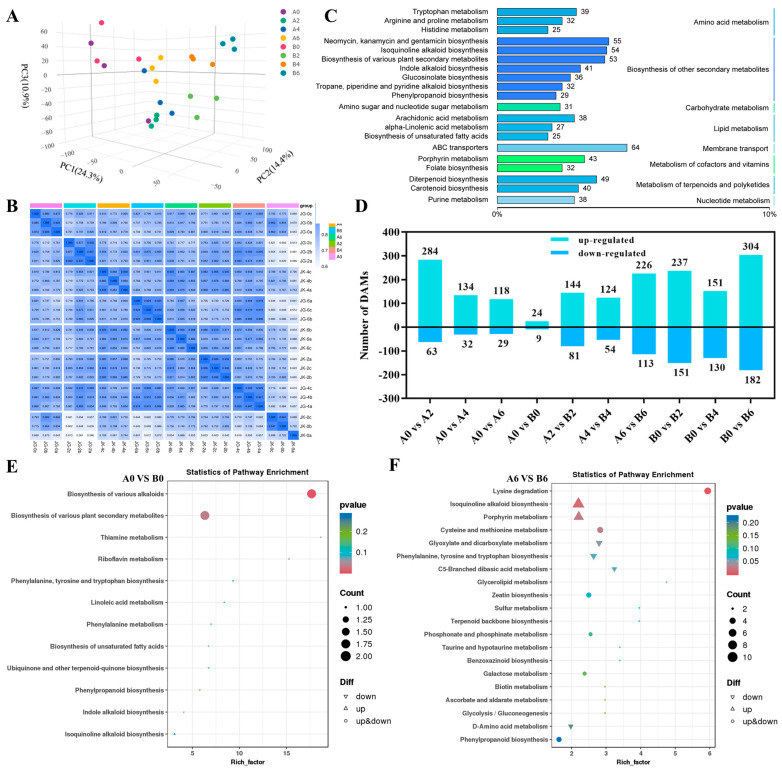
Analysis of metabolome differences between Mc−4 and Mc−18 under FOC stress conditions. (**A**) PCA of DAMs. (**B**) Correlation analysis of DAMs. (**C**) The DAMs in Mc−4 and Mc−18. The length of the bars in the bar chart and the numbers on the right side represent the number of DAMs between groups. (**D**) Changes in DAMs in the Mc−4 and Mc−18 under FOC stress. “Up” represents upregulated metabolites, and “down” represents downregulated metabolites. (**E**) KEGG pathway of A0 vs. B0 differential metabolites. (**F**) KEGG pathway of A6 vs. B6 differential metabolites. A0, and B0: Mc−4 and Mc−18, which were not inoculated with FOC, served as the control group. A2, A4, and A6: Mc−4 was treated with FOC for 2, 4, and 6 days. B2, B4, and B6: Mc−18 was treated with FOC for 2, 4, and 6 days. The darker color and larger size of the triangles and circles in the figure indicate that the corresponding metabolic pathway plays an important role.

**Figure 5 plants-14-00359-f005:**
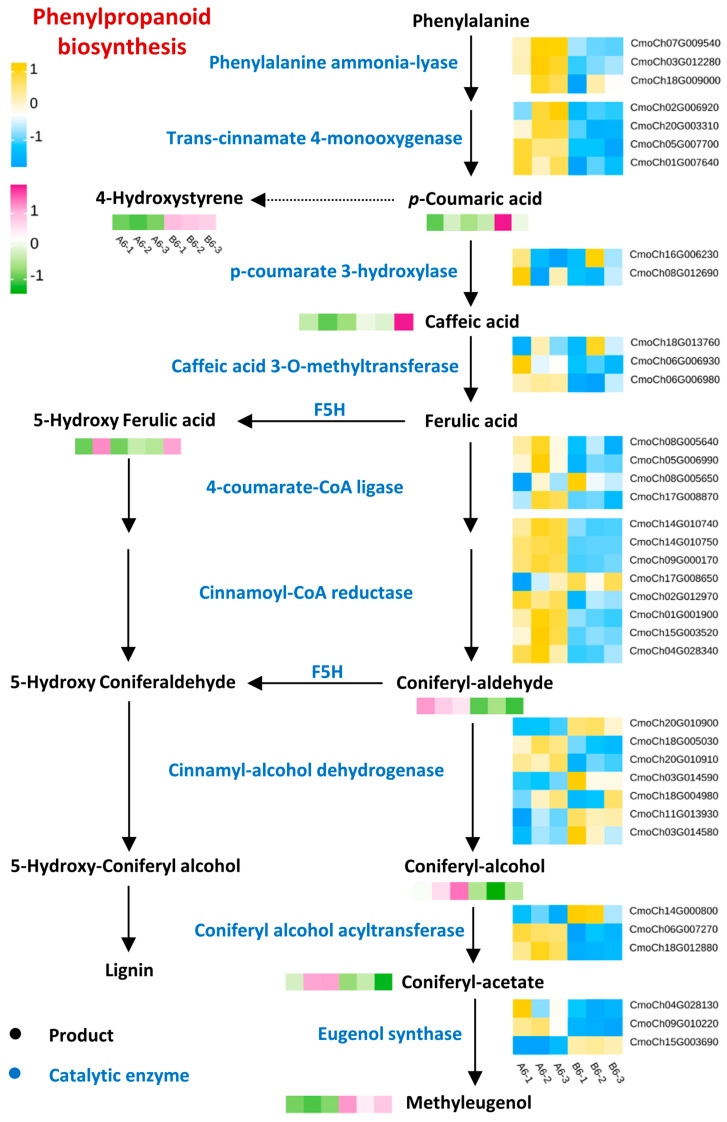
Phenylpropanoid biosynthesis of A6, B6 in Mc−4 and Mc−18 under FOC stress.

**Figure 6 plants-14-00359-f006:**
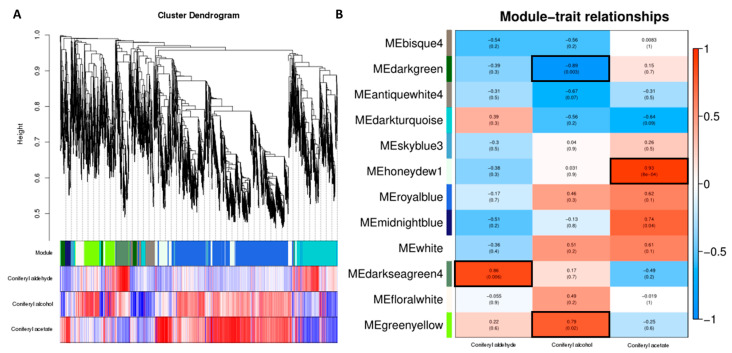
WGCNA analysis of Mc−4 and Mc−18 under FOC stress. (**A**) Hierarchical cluster trees show the co-expression modules identified by WGCNA. (**B**) Co-expression modules by WGCNA. Relationships between modules (left) and traits (bottom). Red and blue represent positive and negative correlations, respectively, with coefficient values and *p*-values.

**Table 1 plants-14-00359-t001:** Incidence rate, DSI, disease resistance level, and AUDRC for different rootstocks inoculated with FOC after 21 days.

Material Code	Cultivars	IR (%)	DSI	RDL	AUDRC (%)
Mc-CK	QuanFuTaiLang	100	82.5 ± 5.4 ^f^	HS	20.6 ± 2.5 ^gh^
Mc-1	QinGenZhenWang	76.7	40.8 ± 3.7 ^b^	MR	16.6 ± 0.1 ^e^
Mc-2	RiBenXueSong	80	40.8 ± 3.3 ^b^	MR	11.9 ± 0.1 ^b^
Mc-3	ShuangLiTieJia	63.3	32.5 ± 1.8 ^a^	R	11.5 ± 0.3 ^b^
Mc-4	JinJiaZhen	60	30.0 ± 3.5 ^a^	R	7.7 ± 0.8 ^a^
Mc-5	ChuangFan No.1	70	28.3 ± 3.6 ^a^	R	15.0 ± 0.3 ^d^
Mc-6	ChaoFan No.1	96.7	66.7 ± 8.0 ^de^	S	20.7 ± 2.0 ^gh^
Mc-7	JinLi No.1	93.3	40.8 ± 0.9 ^b^	MR	12.8 ± 0.8 ^c^
Mc-8	HanGuoJinZhen	83.3	40.0 ± 4.9 ^b^	MR	11.1 ± 1.5 ^b^
Mc-9	JinXinZhen No.5	96.7	54.2 ± 6.9 ^c^	MR	14.9 ± 1.4 ^d^
Mc-10	RiXiDongZhiZhu	100	65.0 ± 6.8 ^d^	S	25.7 ± 0.1 ^i^
Mc-11	QinTengTaiMu	93.3	68.3 ± 7.0 ^de^	S	23.1 ± 3.2 ^h^
Mc-12	ChuangFan No.8	96.7	79.2 ± 9.0 ^ef^	HS	28.8 ± 3.2 ^j^
Mc-13	GenWang 99	100	72.5 ± 7.1 ^e^	S	20.6 ± 1.2 ^gh^
Mc-14	BoQiang 208	93.3	72.5 ± 1.9 ^e^	S	19.4 ± 2.1 ^fg^
Mc-15	ChuangFan 301	93.3	72.5 ± 3.0 ^e^	S	21.6 ± 0.1 ^h^
Mc-16	FengHuang No.3	53.3	50.8 ± 0.4 ^c^	MR	18.8 ± 0.3 ^f^
Mc-17	QiangLiShi	90	71.7 ± 5.7 ^e^	S	18.0 ± 2.0 ^f^
Mc-18	JinGangZhuan 1901	96.7	80.8 ± 3.8 ^f^	HS	22.8 ± 3.3 ^h^
Mc-19	GenLiShen	100	81.7 ± 3.3 ^f^	HS	21.3 ± 1.0 ^h^
Mc-20	HaiZhen No.1	100	74.2 ± 0.4 ^e^	S	22.4 ± 2.6 ^h^

Note: “±” indicates the standard deviation; different letters following the data in the same column denote significant differences (*p* < 0.05). According to reference [11], the level of rootstock resistance to disease based on the DSI can be categorized into 5 classes: highly resistant (HR), resistant (R), moderately resistant (MR), susceptible (S), and highly susceptible (HS). The disease severity index is assigned based on the extent of visible symptoms on plant tissues, expressed as a percentage to quantify the degree of infection. More details on index calculation can be found in the Materials and Methods section.

## Data Availability

All data generated or analyzed during this study are included in this published article.

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
