# Peer review of "Rootstock Selection for Resisting Cucumber *Fusarium* Wilt in Hainan and Corresponding Transcriptome and Metabolome Analysis"

_plants, 2025, doi:10.3390/plants14030359_

Round 1

Reviewer 1 Report

Comments and Suggestions for Authors

This manuscript described transcriptomic and metabolomic comparison of lines differing in susceptibility to Fusarium and identifies phenylpropanoids as a significant difference that may contribute to resistance.  The manuscript is generally well-written, however, there are significant concerns as described below.

1. Critical information is missing about the plant material, fungal inoculum, and growth conditions.

a.  Which plant species is/are being tested (if Cucurbita, which Cucurbita species)?  What was the basis for choosing the lines that were tested?  Have they been used as rootstocks in other work?  If so, this should be clarified. 

b. How and where were the plants grown and under what conditions (soil, temperature, humidity, lighting, irrigation)? Growth chamber, greenhouse, field?  How many replicates were grown of each line for each experiment?

c. How were the Fusarium isolates grown and how was the inoculum prepared?  Which isolates were used/where were they collected from? What was the concentration of inoculum used for the experiments?  How was the inoculum applied?  Directly to the plant? How? Into the soil? How much?  When/at what stage of plant development?

d. how were roots sampled for transcriptomic and metabolomic analyses?  How were they removed from the soil?  What portion of the roots were sampled?

 2. Focus on ‘validation in Hainan’ relative to other locations.  The manuscript refers to validation in Hainan several times.  However, unless the rootstocks were tested in field conditions in replicated trials, it is not clear how the data presented here have value for that question (from the photographs provided, it seems likely that the plants were grown in a growth chamber or greenhouse).  Is there reason to anticipate that rootstocks that were useful in other locations would not be useful in Hainan?  If so, cite relevant references or provide data.  Alternatively, were the Fusarium isolates used in this study unique to Hainan - are there criteria to indicate they are distinct pathotypes? 

3. Lack of reference to relevant recent literature.  There have been several studies in recent years that have specifically looked at transcriptome and/or metabolome responses to pathogen infection in cucumber.  They, like this work, also described increased expression of phenylpropanoid and related pathways, but were not referenced in the manuscript. Some examples are provided below. 

Wang YD et al 2023. Transcriptome Analyses Revealed the Wax and Phenylpropanoid Biosynthesis Pathways Related to Disease Resistance in Rootstock-Grafted Cucumber. Plants. DOI10.3390/plants12162963

Sun YH et al 2023. The reverse mutation of CsMLO8 results in susceptibility to powdery mildew via inhibiting cell wall apposition formation and cell death in cucumber (Cucumis sativus L.) Scientia Horticulturae

Ren et al. 2023. Integrated metabolome and transcriptome analysis provides new insights into the glossy graft cucumber fruit (Cucumis sativus L.). International Journal of Molecular Science

Mansfeld BN et al. 2020. Developmentally regulated activation of defense allows for rapid inhibition of infection in age-related resistance to Phytophthora capsici in cucumber fruit. BMC Genomics

Sa RN et al. 2020. Transcriptome analysis of mechanisms and candidate genes associated with cucumber response to cucumber alternaria leaf spot infection. Physiological and Molecular Plant Pathology

Mansfeld BN et at. 2017. Transcriptomic and metabolomic analyses of cucumber fruit peels reveal a developmental increase in terpenoid glycosides associated with age-related resistance to Phytophthora capsici. Horticulture Research

4.  Information should be provided about the timeline for disease progression.  How long after inoculation did symptoms appear?  How does this relate to the choice of day 6 for transcriptomic and metabolomic analyses?  Related to this, for Figure 2, what does the X-axis mean?  The x-axis should indicate plant age, dates, or days after inoculation.  It would also be valuable to include at least one susceptible line for comparison.

5. line 187-188. “All identified metabolites were annotated using the KEGG database, primarily annotating to biological pathways such as neomycin, kanamycin, and gentamicin biosynthesis (55 species)...”  As plants do not make these compounds, this raises questions about the methods for identification of metabolites.

Other notes:

2. RLD.  Please define in Table 1 and the methods section how the disease rating categories were assigned:  based on which data set (e.g., DSI)?  What levels were the cut off points for each rating?  (note: the disease rating term should be ‘disease resistance level’, not ‘resist disease level’)

3. Figure 3 E,F,I,J,K,L.  These figures should indicate significance level for the different categories.

4. Figure 4E, count legend.  Does this legend indicate fractional values for number of metabolites (i.e., 1.25 metabolites?).   Was the number of metabolites for the categories shown in the figure only 1 or 2?  Also, the P values were very high, raising questions about the level of significance.

5. line 398. What was the basis for the DSI (disease severity index) rating? How were the different levels defined?  It would be helpful to describe the different levels and/or show photographs.

Minor items:

1. line 100.  Should Mc-4 be Mc-5?

2. line 102-103.  It was not clear what is meant by ‘dominant varieties’?  is this a genetic designation?  Based on what analysis?  (also typos in line 103: drisease ledvel)

3. line 106.  ‘Mc-4 showing the best performance’.  Based on what criteria? How does this sentence differ from the prior sentence?

4. line128. Aligned to a reference genome?  Which one?

5. line 132. Define A0, A2, etc. 

6. line 247-248.  The meaning of this sentence was not clear.  Should it be part of the methods?

7. line 249. Why was it of interest to identify genes with low abundance?  What is meant by a low abundance gene?

8. line 390. Bacterial??

9. line 392. What indices?

10. line 402.  Plants are not ‘pathogenic’.  Symptomatic?

11. line 403. ‘morbidity’ is not an appropriate term. ‘symptom severity’?

Author Response

Comments 1: Critical information is missing about the plant material, fungal inoculum, and growth conditions.

  1. Which plant species is/are being tested (if Cucurbita, which Cucurbita species)? What was the basis for choosing the lines that were tested? Have they been used as rootstocks in other work? If so, this should be clarified.
  2. How and where were the plants grown and under what conditions (soil, temperature, humidity, lighting, irrigation)? Growth chamber, greenhouse, field? How many replicates were grown of each line for each experiment?
  3. How were the Fusarium isolates grown and how was the inoculum prepared? Which isolates were used/where were they collected from? What was the concentration of inoculum used for the experiments? How was the inoculum applied? Directly to the plant? How? Into the soil? How much? When/at what stage of plant development?
  4. How were roots sampled for transcriptomic and metabolomic analyses? How were they removed from the soil? What portion of the roots were sampled?

Response 1: Thank you very much for your input. We have provided detailed descriptions for all the issues. The specific content can be found on page 13, lines 407-426 of the manuscript.

Comments 2: Focus on ‘validation in Hainan’ relative to other locations. The manuscript refers to validation in Hainan several times. However, unless the rootstocks were tested in field conditions in replicated trials, it is not clear how the data presented here have value for that question (from the photographs provided, it seems likely that the plants were grown in a growth chamber or greenhouse). Is there reason to anticipate that rootstocks that were useful in other locations would not be useful in Hainan? If so, cite relevant references or provide data. Alternatively, were the Fusarium isolates used in this study unique to Hainan - are there criteria to indicate they are distinct pathotypes?

Response 2: Thank you for your comments. Hainan, as the southernmost island province of China, characterized by high temperatures and humidity, serves as a vital region for studying cucumber fusarium wilt. Furthermore, we wish to clarify that the field trials were conducted under natural environmental conditions in Hainan, rather than in controlled greenhouses or growth chambers. Your confusion on this matter may have arisen due to our lack of clarity regarding the cultivation conditions of the rootstocks in the Materials and Methods section. We have already made the necessary revisions to address this issue.

Regarding your concern about the suitability of rootstocks in Hainan, we have conducted resistance experiments to screen for the most suitable resistant rootstocks for the Hainan region. The resistance of these rootstocks in other regions is not our primary focus. Furthermore, we used Fusarium isolates that were collected specifically from Hainan. This was done to ensure that our findings would be relevant and applicable to the unique conditions in Hainan.

Comments 3: Lack of reference to relevant recent literature. There have been several studies in recent years that have specifically looked at transcriptome and/or metabolome responses to pathogen infection in cucumber. They, like this work, also described increased expression of phenylpropanoid and related pathways, but were not referenced in the manuscript. Some examples are provided below.

Wang YD et al 2023. Transcriptome Analyses Revealed the Wax and Phenylpropanoid Biosynthesis Pathways Related to Disease Resistance in Rootstock-Grafted Cucumber. Plants. DOI10.3390/plants12162963

Sun YH et al 2023. The reverse mutation of CsMLO8 results in susceptibility to powdery mildew via inhibiting cell wall apposition formation and cell death in cucumber (Cucumis sativus L.) Scientia Horticulturae

Ren et al. 2023. Integrated metabolome and transcriptome analysis provides new insights into the glossy graft cucumber fruit (Cucumis sativus L.). International Journal of Molecular Science

Mansfeld BN et al. 2020. Developmentally regulated activation of defense allows for rapid inhibition of infection in age-related resistance to Phytophthora capsici in cucumber fruit. BMC Genomics

Sa RN et al. 2020. Transcriptome analysis of mechanisms and candidate genes associated with cucumber response to cucumber alternaria leaf spot infection. Physiological and Molecular Plant Pathology

Mansfeld BN et at. 2017. Transcriptomic and metabolomic analyses of cucumber fruit peels reveal a developmental increase in terpenoid glycosides associated with age-related resistance to Phytophthora capsici. Horticulture Research

Response 3: Thank you for your comments, mentioned papers have already cited in the manuscript.

Comments 4: Information should be provided about the timeline for disease progression. How long after inoculation did symptoms appear? How does this relate to the choice of day 6 for transcriptomic and metabolomic analyses?

Related to this, for Figure 2, what does the X-axis mean? The x-axis should indicate plant age, dates, or days after inoculation. It would also be valuable to include at least one susceptible line for comparison.

Response 4: Thank you for your comments, we agree with your comments. The indexes summary which represent rootstocks’ disease progression was shown in Table S6 in Supporting Material.

The X-axis in Figure 2 represents the time elapsed since the inoculation of the rootstocks. The 21st day of the experiment is normalized to 1 on the X-axis, while the first day is normalized to 0. For reference, the 5th day is represented as 0.2 on the X-axis. This time transformation was introduced to align with the definition of the Area Under the Receiver Operating Characteristic (AUC) curve in machine learning, ensuring that the highest AUC value is standardized to 1. This normalization will facilitate future comparisons between different experiments (Page 4, lines 117-124).

Comments 5: line 187-188. “All identified metabolites were annotated using the KEGG database, primarily annotating to biological pathways such as neomycin, kanamycin, and gentamicin biosynthesis (55 species)...” As plants do not make these compounds, this raises questions about the methods for identification of metabolites.

Response: Thank you for your comment. We have double checked the results of Metabolome analysis and manuscript, there are no typos in this sentence.

Regarding the attendance of pathway ‘neomycin, kanamycin, and gentamicin biosynthesis’ in our Metabolome analysis, this phenomenon may indicate a potential involvement of plant defense mechanisms. Neomycin, kanamycin, and gentamicin are aminoglycoside antibiotics, and their biosynthesis involves intermediate metabolites that could play roles in antimicrobial defense in plants. More specific, during pathogen infection, plants may activate secondary metabolism pathways that produce antimicrobial compounds, either directly inhibiting pathogen growth or enhancing overall resistance. The metabolites related to the biosynthesis of neomycin, kanamycin, and gentamicin could serve similar roles in plant defense. Similar phenomenon was reported in some published studies [DOI: 10.1038/35081178; 10.3390/molecules23040762].

Comments 6: RLD. Please define in Table 1 and the methods section how the disease rating categories were assigned: based on which data set (e.g., DSI)? What levels were the cut off points for each rating? (note: the disease rating term should be ‘disease resistance level’, not ‘resist disease level’)

Response 6: Thank you for your comments, we agree with your comments. We have provided a detailed description on page 13, lines 436-440 of the manuscript.

Comments 7: Figure 3 E, F, I, J, K, L. These figures should indicate significance level for the different categories.

Response 7: Thank you for your comments, we agree with your comments. These figures only present the statistical count of DEGs across different pathways, without performing differential analysis between different groups. A longer bar in the histogram indicates a higher number of enriched DEGs.

Comments 8: Figure 4E, count legend. Does this legend indicate fractional values for number of metabolites (i.e., 1.25 metabolites?). Was the number of metabolites for the categories shown in the figure only 1 or 2? Also, the P values were very high, raising questions about the level of significance.

Response 8: Thank you for your comments, we agree with your comments. The size of the circles represents the number of DAMs, with the largest circle indicating 2 DAMs. There will not be a situation with 1.25 metabolites, suggesting that there are fewer DAMs between A0 and B0. In Figure 4E, except for the two larger circles with P-values below 0.05, the P-values for the remaining pathways are relatively high. This may be due to the fewer DAMs between A0 VS B0, further indicating that there is no significant difference in DAMs between A0 VS B0. This indirectly suggests that the DAMs between A6 VS B6 are caused by FOC.

Comments 9: line 398. What was the basis for the DSI (disease severity index) rating? How were the different levels defined? It would be helpful to describe the different levels and/or show photographs.

Response 9: Thank you for your comments, we agree with your comments. We have provided a detailed description on page 13, lines 436-440 of the manuscript.

Comments 10: line 100. Should Mc-4 be Mc-5?

Response 10: Thank you for your comments, we agree with your comments. We have changed Mc-4 to Mc-5 in the article (Page 3, line 93).

Comments 11: line 102-103. It was not clear what is meant by ‘dominant varieties’? is this a genetic designation? Based on what analysis? (also typos in line 103: drisease ledvel)

Response 11: Thank you for your comments, we agree with your comments. ‘dominant varieties’ was corrected as ‘better rootstock’ and typos have been corrected (Page 3, lines 95-96).

Comments 12: line 106. ‘Mc-4 showing the best performance’. Based on what criteria? How does this sentence differ from the prior sentence?

Response 12: Thank you for your comments, we agree with your comments. Corresponding content has been corrected as ‘Mc-4 showing the best performance for three mentioned resistance (R) rootstocks (One-way anova analysis of AUDRC, p = 0.02)’ (Page 3, lines 98-101).

Comments 13: line128. Aligned to a reference genome? Which one?

Response 13: I fully agree with your suggestion, and we have provided a detailed description of the reference genome in Table S1. Reference genome: Cucurbita moschata (Rifu) (http://cucurbitgenomics.org/organism/9).

Comments 14: line 132. Define A0, A2, etc.

Response 14: Thank you for your comments, we agree with your comments.A0 and A2 means samples were collected from MC-4 at 0 days, 2 days post-inoculation with FOC (Page 5, lines 140-141, 156-157).

Comments 15: line 247-248. The meaning of this sentence was not clear. Should it be part of the methods?

Response 15: I fully agree with your suggestion. This part overlaps with the Materials and Methods section, so we have made deletions accordingly (Page 9, lines 262--264).

Comments 16: line 249. Why was it of interest to identify genes with low abundance? What is meant by a low abundance gene?

Response 16: I apologize for the confusion earlier. We have corrected the misuse of terminology and intended to express ‘high abundance and low variability genes’ (Page 9, lines 262--264).

Comments 17: line 390. Bacterial??

Response 17: Thank you for your comments, we agree with your comments. We have changed bacterial to fungus solution in the article (Page 13, lines 407--426).

Comments 18: line 392. What indices?

Response 18: Thank you very much for your feedback. The physiological indicators used as a reference for rootstock selection are introduced in detail in the subsequent Materials and Methods section. To avoid potential ambiguities here, we have revised the description accordingly (Page 13, lines 407--426).

Comments 19: line 402. Plants are not ‘pathogenic’. Symptomatic?

Response 19: Your suggestion is correct, and we have made the necessary changes (Page 13, line 434).

Comments 20: line 403. ‘morbidity’ is not an appropriate term. ‘symptom severity’?

Response 20: Your suggestion is correct, and we have made the necessary changes (Page 13, line 435).

Reviewer 2 Report

Comments and Suggestions for Authors

1.  In the research objective section, the limitations of previous studies on the selection of cucumber fusarium wilt-resistant rootstocks are not clearly elaborated. For example, it only mentions that most studies lack verification in Hainan, but fails to deeply analyze the specific impact of this limitation on the current research and the reasons why this research needs to be carried out in Hainan.

2.   In the section of plant materials and rootstock selection, the description of the sources of 21 rootstock varieties is not detailed enough. Only the companies they are from are mentioned, without information such as the genetic background and previous resistance performance of these varieties, making it difficult for readers to evaluate the representativeness and diversity of the selected varieties.

3.   In the transcriptome and metabolome analysis, although a series of analyses were carried out on differentially expressed genes (DEGs) and differentially accumulated metabolites (DAMs), the statistical analysis of experimental repeatability should be added. For example, calculate the coefficient of variation (CV) among replicate samples to show the degree of consistency of transcriptome and metabolome data in different replicates.

4.   The titles and axis labels of some figures (such as Figure 1, Figure 3, Figure 4, etc.) are not detailed enough, making it difficult for readers to directly obtain key information from the figures. For example, Figure 1 only labels the difference diagram of rootstocks 21 days after inoculation with FOC, but does not explain in which aspects these differences are specifically reflected (such as plant growth status, disease symptoms, etc.).

5.   There are some grammatical errors and inappropriate word usages in the paper, which affect the readability of the paper. For example, in the sentence "Rootstock selection for grafting to avoid soil - borne disease is a well - studied and widespread challenge.", the position of "to avoid soil - borne disease" is not appropriate, which is likely to cause ambiguity in understanding.

6.   When describing experimental materials and methods, the use of some professional terms is not consistent. For example, "Fusarium oxysporum f. sp. Cucumerinum" is sometimes abbreviated as "FOC" and sometimes written in full in the paper, which is likely to cause confusion.

7.   Literature: The citation of important classic literatures is lacking. For example, the citations of "A chitinase CsChi23 promoter polymorphism underlies cucumber resistance against Fusarium oxysporum f. sp. Cucumerinum" and "Comprehensive Analysis of the Chitinase Gene Family in Cucumber (Cucumis sativus L.): From Gene Identification and Evolution to Expression in Response to Fusarium oxysporum" are missing. At the same time, the reference formats are inconsistent. For example, for references 32 and 36, it should be the first name that is abbreviated instead of the surname, and it should be consistent with references 21 and 20.

Comments on the Quality of English Language

No suggestions, need to be improved by native speaker.

Author Response

Comments 1: In the research objective section, the limitations of previous studies on the selection of cucumber fusarium wilt-resistant rootstocks are not clearly elaborated. For example, it only mentions that most studies lack verification in Hainan, but fails to deeply analyze the specific impact of this limitation on the current research and the reasons why this research needs to be carried out in Hainan.

Response 1: Thank you for pointing this out. We agree with this comment. Therefore, we have provided a detailed description of the limitations of previous research on the selection and breeding of cucumber rootstocks resistant to Fusarium wilt. The specific modifications are located on lines 51-61 of the second page of the manuscript.

Comments 2: In the section of plant materials and rootstock selection, the description of the sources of 21 rootstock varieties is not detailed enough. Only the companies they are from are mentioned, without information such as the genetic background and previous resistance performance of these varieties, making it difficult for readers to evaluate the representativeness and diversity of the selected varieties.

Response 2: Thank you very much for your input. All of these 21 rootstocks are derived from pumpkin. The primary reasons for selecting pumpkin rootstocks are as follows: (1) Pumpkin rootstocks exhibit significant resistance to diseases such as watermelon Fusarium wilt; (2) Pumpkin rootstocks possess strong adaptability and can grow in various soil and climatic conditions; (3) Pumpkin rootstocks have vigorous root systems and high disease resistance, which often lead to higher yields in grafted crops. Furthermore, we have conducted experiments to verify the resistance performance of these 21 rootstocks, and the results can be found in Table 1. Additional information on the genetic backgrounds of these 21 rootstocks is provided on lines 407-426 of page 13 of the manuscript.

Comments 3: In the transcriptome and metabolome analysis, although a series of analyses were carried out on differentially expressed genes (DEGs) and differentially accumulated metabolites (DAMs), the statistical analysis of experimental repeatability should be added. For example, calculate the coefficient of variation (CV) among replicate samples to show the degree of consistency of transcriptome and metabolome data in different replicates.

Response 3: Thank you very much for your advice and suggestions. We have completed the calculation of the coefficient of variation (CV). The coefficient of variation results between replicate samples for DEGs (Differentially Expressed Genes) and DAMs (Differentially Accumulated Metabolites) indicate a good degree of consistency in the data across different replicates (Table S2). We have made the modifications on page 7, lines 230-232 of the manuscript.

Comments 4: The titles and axis labels of some figures (such as Figure 1, Figure 3, Figure 4, etc.) are not detailed enough, making it difficult for readers to directly obtain key information from the figures. For example, Figure 1 only labels the difference diagram of rootstocks 21 days after inoculation with FOC, but does not explain in which aspects these differences are specifically reflected (such as plant growth status, disease symptoms, etc.).

Response 4: In response to your questions, we have enriched the figure captions to ensure that key information can be obtained from the charts and tables as much as possible (Page 3, lines 111-116; Page 4, lines 118-124; Page 6, lines 189-192; Page 8, lines 250-258).

Comments 5: There are some grammatical errors and inappropriate word usages in the paper, which affect the readability of the paper. For example, in the sentence "Rootstock selection for grafting to avoid soil - borne disease is a well - studied and widespread challenge.", the position of "to avoid soil - borne disease" is not appropriate, which is likely to cause ambiguity in understanding.

Response 5: Thank you very much for your suggestions. The manuscript has undergone a comprehensive review and revision by a professional whose native language is English.

Comments 6: When describing experimental materials and methods, the use of some professional terms is not consistent. For example, "Fusarium oxysporum f. sp. Cucumerinum" is sometimes abbreviated as "FOC" and sometimes written in full in the paper, which is likely to cause confusion.

Response 6: We strongly concur with your opinion and have made comprehensive adjustments to address this issue in order to prevent any confusion. The entire text retains "Fusarium oxysporum f. sp. Cucumerinum" only in the abstract and in line 80 on page 3, with "FOC" used elsewhere. We have also adjusted all other abbreviations.

Comments 7: Literature: The citation of important classic literatures is lacking. For example, the citations of "A chitinase CsChi23 promoter polymorphism underlies cucumber resistance against Fusarium oxysporum f. sp. Cucumerinum" and "Comprehensive Analysis of the Chitinase Gene Family in Cucumber (Cucumis sativus L.): From Gene Identification and Evolution to Expression in Response to Fusarium oxysporum" are missing. At the same time, the reference formats are inconsistent. For example, for references 32 and 36, it should be the first name that is abbreviated instead of the surname, and it should be consistent with references 21 and 20.

Response 7: Thank you very much for your feedback. We have supplemented the references and thoroughly checked the citation format.

Round 2

Reviewer 1 Report

Comments and Suggestions for Authors

The revised manuscript is much improved.  There are still a few items to be considered:

1. Methods.  Thank you for the additional information, it is very helpful.  Some remaining questions:

a. what was the basis for selecting the specific rootstocks chosen?  Perhaps it is because they are commercially available?  If so, it would help to state this, i.e., “In this study, a total of 21 commercial rootstock varieties derived from Cucurbita moschata...”  or were they chosen because they were the ones used in reference 47? If so, that should be stated, i.e., ‘“In this study, a total of 21 rootstock varieties derived from Cucurbita moschata as described by Yang [47]...”

b. how is a replicate defined?  What was the planting design in the field (i.e., randomized complete block?  Number of plants per replicate? Spacing between plants…)? 

c. What was the soil type?  This is relevant especially since the roots are being inoculated.

d. line 414. What was the ‘preparatory test’?  define for reader.

e. line 416-418. Bacteria?? Again?  It is disturbing to see this level of carelessness.

f. line 417-418.  The inoculum was purified?  How?  How was it cultured to provide sufficient quantities needed for inoculation?   

g. line 419.  What is ‘heart’?’

h. line 420. ‘human injury’??

i. is it correct to understand that people went into the field and hand inoculated roots in field plots? (although not impossible, seems difficult to be sure to properly reach the root) Or was inoculation done prior to transplanting?  Were seedlings first germinated in the greenhouse?

j. line 425. The plants were ‘inoculated’ not ‘vaccinated’

2. According to the methods, the plants were inoculated at the 2-leaf stage (line 419), then observed for 21 days.  In the photographs in Fig 1, the plants only have 3 leaves (even for the resistant line #4).  This lack of growth over a three-week period is very surprising.

3. Figure 4E. Given that there were at most only two metabolites, it does not seem appropriate to include this figure.  The statement (lines 215-217), “In the comparison of A0 vs. B0, the DAMs are primarily enriched in the biosynthesis of various alkaloids and the biosynthesis of various plant secondary metabolites (Figure 4E)” seems to be an overstatement for two compounds.

Some additional notes:

1. The species for the rootstocks should also be clearly stated in the introduction.

2 Figure 1.  It would be helpful to the reader when referring to a line by name, to also include the number so it can be found more easily.  Also, note that ‘remarkable disease resistance’ is not evident from the figure; this phrase should be deleted.  Also note that Fig 1 is photographs, not diagrams (see both figure legend and line 88).

3. Table 1.  It is very helpful that the basis for assigning disease severity index was included in the methods. It also should be included in the table footnotes.

4. line 421-423. This sentence is not in proper format, please revise.

5. While some of the mentioned papers to be considered were included in the revised draft, one was cited as relevant to grafting when there was no grafting work (ref 8 in line 40) while several of the other papers that had been listed (e.g., Wang, Ren) do deal with grafting, as well as numerous uncited papers dealing more generally with effects of grafting in cucumber (some additional examples are listed below).  On the other hand, ref 8 deals with phenylpropanoid related pathways and pathogen defense but was not cited in that context. 

Integrated Metabolome and Transcriptome Analysis Provide Insights into the Effects of Grafting on Fruit Flavor of Cucumber with Different Rootstocks. Miao L, Di Q, Sun T, Li Y, Duan Y, Wang J, Yan Y, He C, Wang C, Yu X. Int J Mol Sci. 2019 Jul 23;20(14):3592. doi: 10.3390/ijms20143592.

Humic acid and grafting as sustainable agronomic practices for increased growth and secondary metabolism in cucumber subjected to salt stress.  Amerian M, Palangi A, Gohari G, Ntatsi G. Sci Rep. 2024 Jul 10;14(1):15883. doi: 10.1038/s41598-024-66677-8.

Cucumber grafting on indigenous cucurbit landraces confers salt tolerance and improves fruit yield by enhancing morpho-physio-biochemical and ionic attributes. Abbas F, Faried HN, Akhtar G, Ullah S, Javed T, Shehzad MA, Ziaf K, Razzaq K, Amin M, Wattoo FM, Hafeez A, Rahimi M, Abeed AHA. Sci Rep. 2023 Dec 7;13(1):21697. doi: 10.1038/s41598-023-48947-z.

Grafting-enhanced tolerance of cucumber to toxic stress is associated with regulation of phenolic and other aromatic acids metabolism. Xiao X, Li J, Lyu J, Hu L, Wu Y, Tang Z, Yu J, Calderón-Urrea A. PeerJ. 2022 Jun 1;10:e13521. doi: 10.7717/peerj.13521. eCollection 2022.

Author Response

Comments 1: Methods. Thank you for the additional information, it is very helpful. Some remaining questions:

  1. what was the basis for selecting the specific rootstocks chosen? Perhaps it is because they are commercially available? If so, it would help to state this, i.e., “In this study, a total of 21 commercial rootstock varieties derived from Cucurbita moschata...” or were they chosen because they were the ones used in reference 47? If so, that should be stated, i.e., ‘“In this study, a total of 21 rootstock varieties derived from Cucurbita moschata as described by Yang [47]...”
  2. how is a replicate defined? What was the planting design in the field (i.e., randomized complete block? Number of plants per replicate? Spacing between plants…)?
  3. What was the soil type? This is relevant especially since the roots are being inoculated.
  4. line 414. What was the ‘preparatory test’? define for reader.
  5. line 416-418. Bacteria?? Again? It is disturbing to see this level of carelessness.
  6. line 417-418. The inoculum was purified? How? How was it cultured to provide sufficient quantities needed for inoculation?
  7. line 419. What is ‘heart’?’
  8. line 420. ‘human injury’??
  9. is it correct to understand that people went into the field and hand inoculated roots in field plots? (although not impossible, seems difficult to be sure to properly reach the root) Or was inoculation done prior to transplanting? Were seedlings first germinated in the greenhouse?
  10. line 425. The plants were ‘inoculated’ not ‘vaccinated’

Response a-e: Thank you for your comments, this section has been added (Lines 423-443).

Response e: Thank you for your comments, we agree with your comments. We have changed Bacterial to fungi in the article (Lines 444-446).

Response f: Thank you for your comments, this section has been added (Lines 430-441).

Response g: Thank you for your comments. The 2 leaves and 1 heart stage refers to the period when the plant has developed two true leaves and a growing apex.

Response h: Sorry for our typos, ‘human injury’ should be corrected as ‘human inoculation’ (Line 448).

Response i: As we mentioned in manuscript, the fungi inoculation finished in following method ’On the premise of being 1-2 cm away from the taproot and not harming the taproot, use a knife to cut down from the surface of the soil to injure the roots of the plant and inject 10 ml of the prepared fungus solution with a syringe.’ the mentioned inoculation method is a typical method for fungi inoculation, therefore, we think your understanding is correct.

Seedlings for our experiment were finished in field.

Response j: Thank you for your comments, We have changed in the article.

Comments 2: According to the methods, the plants were inoculated at the 2-leaf stage (line 419), then observed for 21 days. In the photographs in Fig 1, the plants only have 3 leaves (even for the resistant line #4). This lack of growth over a three-week period is very surprising.

Response 2: Thank you for your comments, the typical image of rootstock in 2-leaf stage before inoculation is shown as follow. Although something surprising was happened, it was recorded by our experiments follow with our designed method. Possible reasons for mentioned phenomenon are show as following: phytomass increasing of pumpkin in starting stage is low [DOI:10.5433/1679-0359.2022v43n3p1211], additional, stress from FOC reduce rootstock’s growth in some degree even push most of rootstocks to decay during 3 weeks’ experiment.

Figure. typical image of rootstock in 2-leaf stage before inoculation

Comments 3: Figure 4E. Given that there were at most only two metabolites, it does not seem appropriate to include this figure. The statement (lines 215-217), “In the comparison of A0 vs. B0, the DAMs are primarily enriched in the biosynthesis of various alkaloids and the biosynthesis of various plant secondary metabolites (Figure 4E)” seems to be an overstatement for two compounds.

Response 3: Thank you very much for your input; this mistake was critical for the manuscript. We have already made the necessary corrections accordingly (Page 7, lines 233-235).

Comments 4: The species for the rootstocks should also be clearly stated in the introduction

Response 4: Thank you for your comments, we agree with your comments. In this study, a total of 21 commercial rootstock varieties derived from Cucurbita moschata were selected as potential germplasm resources. Mentioned section has been added into Introduction section (Lines 93-95).

Comments 5: Figure 1. It would be helpful to the reader when referring to a line by name, to also include the number so it can be found more easily. Also, note that ‘remarkable disease resistance’ is not evident from the figure; this phrase should be deleted. Also note that Fig 1 is photographs, not diagrams (see both figure legend and line 88).

Response 5: Thank you for your comments, We have made revisions to Figure 1. Additional, you mentioned commons have been corrected (Lines 125-128).

Comments 6: Table 1. It is very helpful that the basis for assigning disease severity index was included in the methods. It also should be included in the table footnotes.

Response 6: Thank you for your comments, we agree with your comments. The disease severity index is assigned based on the extent of visible symptoms on plant tissues, expressed as a percentage to quantify the degree of infection. Mentioned statement has been added into manuscript into methods section and table footnotes (Lines 145-147).

Comments 7: line 421-423. This sentence is not in proper format, please revise.

Response 7: Thank you for your comments, we agree with your comments. Corresponding sentence was revised as follow: To ensure the taproot remains intact, make an incision 1-2 cm away from it, cutting down from the soil surface to damage the surrounding roots. Then, using a syringe, inject 10 ml of the prepared fungal solution into the incision.

Comments 8: While some of the mentioned papers to be considered were included in the revised draft, one was cited as relevant to grafting when there was no grafting work (ref 8 in line 40) while several of the other papers that had been listed (e.g., Wang, Ren) do deal with grafting, as well as numerous uncited papers dealing more generally with effects of grafting in cucumber (some additional examples are listed below).  On the other hand, ref 8 deals with phenylpropanoid related pathways and pathogen defense but was not cited in that context.

Integrated Metabolome and Transcriptome Analysis Provide Insights into the Effects of Grafting on Fruit Flavor of Cucumber with Different Rootstocks. Miao L, Di Q, Sun T, Li Y, Duan Y, Wang J, Yan Y, He C, Wang C, Yu X. Int J Mol Sci. 2019 Jul 23;20(14):3592. doi: 10.3390/ijms20143592.

Humic acid and grafting as sustainable agronomic practices for increased growth and secondary metabolism in cucumber subjected to salt stress. Amerian M, Palangi A, Gohari G, Ntatsi G. Sci Rep. 2024 Jul 10;14(1):15883. doi: 10.1038/s41598-024-66677-8.

Cucumber grafting on indigenous cucurbit landraces confers salt tolerance and improves fruit yield by enhancing morpho-physio-biochemical and ionic attributes. Abbas F, Faried HN, Akhtar G, Ullah S, Javed T, Shehzad MA, Ziaf K, Razzaq K, Amin M, Wattoo FM, Hafeez A, Rahimi M, Abeed AHA. Sci Rep. 2023 Dec 7;13(1):21697. doi: 10.1038/s41598-023-48947-z.

Grafting-enhanced tolerance of cucumber to toxic stress is associated with regulation of phenolic and other aromatic acids metabolism. Xiao X, Li J, Lyu J, Hu L, Wu Y, Tang Z, Yu J, Calderón-Urrea A. PeerJ. 2022 Jun 1;10:e13521. doi: 10.7717/peerj.13521. eCollection 2022.

Response 8: Thank you for your comments, we agree with your comments. We have revised the position of reference 8 and added new references.

Reviewer 2 Report

Comments and Suggestions for Authors

  The author has made excellent modifications. There are no other suggestions for this version at present.      

Comments on the Quality of English Language

  The author has made excellent modifications. There are no other suggestions for this version at present.       

Author Response

Thank you very much for reviewing our paper and providing valuable feedback. We are thrilled to learn that our manuscript has been accepted. Your affirmation is a great encouragement to our work and has further boosted our confidence in the quality of the paper. We fully embrace your comments and will continue to strive in our subsequent work, constantly enhancing our research capabilities and the quality of our papers.